# An Advanced Data Processing Algorithm for Extraction of Polarimetric Radar Signatures of Moving Automotive Vehicles Using the $H/A/\alpha$ Decomposition Technique

Detmer A. Bosma *, Oleg A. Krasnov and Alexander Yarovoy





Microwave Sensing, Signals & Systems (MS3), Faculty of Electrical Engineering, Computer Science and Mathematics, Delft University of Technology, Mekelweg 4, 2628 CD Delft, The Netherlands
* Correspondence: d.a.bosma@tudelft.nl

**Abstract:** A dedicated signal and data processing chain is proposed for a fully polarimetric Doppler surveillance S-band radar to extract the polarimetric signatures of moving targets. To extract the target's polarimetric features, detection, clustering, and tracking steps are realized for a multi-target environment in the range-Doppler domain. A dedicated data fusion method for all four polarimetric radar channel signals is implemented to take full advantage of the additional polarimetric information and improve the detection performance. While tracking each particular target, polarization information is collected and used to describe their polarization scattering characteristics. Using the polarimetric $H/A/\alpha$ decomposition technique, the polarimetric features of moving automotive targets are extracted and investigated. The developed processing chain has been applied to the signals scattered from vehicles moving in a highway. By employing both time averaging and spatial averaging of the statistical coherency matrix, the polarimetric signatures of both moving vehicles and static clutter have been presented in the two-dimensional $H/\alpha$ plane. It has been found that the spatial averaging approach results in polarimetric signatures of moving vehicles that give the opportunity to directly and without consideration of the motion of the targets compare the polarization features of moving targets and static clutter. Therefore, this method can be used to improve the performance of target detection or target classification.

**Keywords:** polarimetric radar; polarimetric signature; $H/A/\alpha$ decomposition; polarimetric fusion; target detection; multi-target tracking

## 1. Introduction

While radar technology was mainly developed for military and surveillance applications during the 20th century [1], it currently plays a significant role as a sensor in remote sensing (e.g., air and road traffic control) [2]. Nowadays, many practical radar applications require an automatic interpretation of the received data, including data-processing algorithms for automatic tracking and target classification. Many target classification algorithms are lacking due to a shortage of labeled data sets and information about the characteristics of a particular target type, also known as its signature. These signatures are based, for instance, on polarimetric characteristics and can be used to distinguish different types of objects (i.e., different classes). Radars capable of measuring these polarimetric characteristics provide valuable additional information for more reliable target detection, more accurate target identification, and better parameter estimation, as was discussed in [3]. Therefore, the exploitation of polarimetric information is a very promising concept for improving automotive vehicle classification and can be applied in both automotive radar and surveillance radar, such as road traffic control and monitoring [4] or feature-aided tracking of vehicles [5]. To provide this information, the polarimetric signatures of moving automotive vehicles need to be known [6].

In 1960, the first article that described the fundamentals of classification using polarimetric properties was published by Copeland [7]. Many new concepts of exploiting polarimetric information for object identification and classification followed [8,9], stimulated by new theoretical insights into radar polarimetry theory and its relations with target identification problems from, among others, Huynen [10,11] and Boerner [12–15]. According to Huynen's theorem, the dynamic behavior of the scattering matrix can be considered for polarization-based target classification and identification, provided that a complete polarimetric measurement is available [16]. In the following decades, polarimetric radar was used more frequently to improve classification methods, such as the classification of Earth terrain with polarimetric *synthetic aperture radar* (SAR) imaging [17–19], ground-based (military) vehicles [20,21], road surface classification [22–24], and road debris detection [25], which can be used to inform a vehicle driver about critical road situations (e.g., aquaplaning, snow, and icy surfaces).

Due to the increasing interest in automated driving systems, polarization radar became more popular in the automotive sector as well. In order to improve the safety and reliability of these systems, automatic traffic scene interpretation is required, as classification of automotive vehicles is an important job [4]. Therefore, in the past 10 years, a lot of research on the polarimetric features of automotive vehicles has been carried out. For the purpose of having suitable vehicle models for radar simulations, many measurements of the *radar cross-sections* (RCSs) of several automotive vehicles (motor scooters, small cars, estate cars, vans, etc.) using a full polarimetric radar were performed in several frequency bands, both in controlled environments (i.e., in an anechoic chamber) [26–30] and in real-world situations [31]. In particular, Tilly et al., motivated by the work of Visentin [30], researched the polarimetric signatures of passenger cars with real-world measurements from various angles [32]. Based on a random forest classifier, a significant performance improvement was shown by using the additional polarimetric features instead of only using single-polarization scattering information. An analysis of the permutation importance of the polarimetric features verifies that this information indeed contributes to determining the correct classes from some more complex cases [33].

Modern polarimetric waveforms facilitate the ability to measure all four elements of the *polarization scattering matrix* (PSM). These elements (HH, HV, VH, and VV) consist of complex data, meaning that both the amplitude and phase information of the received signals are provided [34]. The four components of the PSM can be decomposed for an easier physical interpretation [35] and can therefore be used to describe the polarimetric signatures of certain classes of objects. Aside from the commonly-used coherent decomposition techniques, such as Pauli and Krogager decomposition, many polarimetric decomposition techniques are based on the coherence matrix, which covers the statistical average of all scattering information, such as the Huynen target parameters, which are related to the geometrical and physical interpretations (orientation, position, shape, etc.), and the $H/A/\alpha$ parameters are derived to describe target scattering [11]. The $H/A/\alpha$ decomposition, based on the eigenvalue decomposition of the coherence matrix, is a typical and useful method for effective feature extraction and is commonly used for the physical interpretation of the scattering mechanism. The entropy $H$ and angle $\alpha$ are often visualized in the two-dimensional $H/\alpha$ plane, which directly presents the type of scattering according to a simple classification scheme [6].

The goal of this research, which is an extension of the work originally presented in [3], is to develop a signal and data processing chain to extract the polarimetric information from multiple moving targets and to study the temporal and spatial variability of this information for different types of automotive targets, highway structures, and other types of objects presented in radar scenes, regardless to their motion. For this reason, we selected the approach to extract the polarimetric information from the observed objects in the range-Doppler domain, using the Doppler velocity as a feature to distinguish different moving targets and (quasi-)stationary objects. These targets are seen by a high-resolution fully polarimetric radar as clusters of detection points in the range-Doppler domain. To obtain the

timeline of the statistical polarimetric features from the targets, determining and tracking of these clusters over time is required. A dedicated signal and data processing chain to process real-world polarimetric data, gathered by the fully polarimetric Doppler surveillance S-band radar PARSAX (see [34,36]), established by the TU Delft, has been developed. A unique feature of the PARSAX radar is its ability to measure all four elements of the PSM quasi-simultaneously using the polarimetric sounding signals with dual orthogonality. This property will be exploited to improve the target detection performance by applying a polarimetric data fusion algorithm before target detection is performed. Subsequently, a *multi-target tracking* (MTT) algorithm is implemented to track each individual vehicle over time. During this process, the polarization information of the clusters corresponding to the moving vehicles is collected and used for the database that describes the polarization-scattering characteristics of these targets. Further statistical processing uses the polarimetric $H/A/\alpha$ decomposition technique to study the difference between the polarimetric features of automotive targets and highway clutter.

The remaining part of the paper is structured as follows. In Section 2, the fully polarimetric Doppler radar PARSAX is introduced, the proposed signal and data processing chain is presented, and the $H/A/\alpha$ decomposition technique is explained. The results of this research are presented in Section 3 and discussed in Section 4. Lastly, the conclusions and recommendations for future research are presented in Section 5.

## 2. Materials and Methods

In this section, an introduction to the fully polarimetric radar PARSAX, developed by the TU Delft, is given, and its radar characteristics are presented. Subsequently, the basic radar signal processing chain is explained and visualized with data captured by PARSAX. Finally, the $H/A/\alpha$ decomposition technique is introduced.

### 2.1. Fully Polarimetric Radar

The PARSAX radar is a fully polarimetric Doppler FMCW surveillance radar developed by the TU Delft (The Netherlands) and originally designed as a weather radar, but it can be utilized for other applications as well. This radar system operates in the S-band (3.315 GHz) with a maximum bandwidth of up to 50 MHz. Since only 90% of the total sweep time data are acquired, the maximum effective bandwidth $B_{eff}$ is 45 MHz. A single sweep has a fixed duration of 1 ms, which is sampled at 400 MHz with a 14 bit resolution. The FPGA-based real-time digital processing makes it possible to use different classes of waveforms and implement complicated algorithms for signal and data processing in real-time with low noise, high sensitivity, and a dynamic range for targets' signals, as well as very weak cross-channel interference levels. An overview of the system and antenna characteristics of the PARSAX radar [36] is provided by Table 1. These characteristics result in an effective range resolution $\Delta R$ of 3.3 m and a radial velocity resolution $\Delta v_r$ of 0.087 m s$^{-1}$, which is equivalent to 0.31 km h$^{-1}$, as the number of sweeps per burst $N_{Sweeps}$ is usually 512 [34]. This is equivalent to an integration time $T_i$ of approximately 0.52 s. Moreover, this radar has a maximum unambiguous radial velocity of $v_r^{max}$ of $\pm 22.1$ m s$^{-1}$, which is equivalent to $\pm 79.8$ km h$^{-1}$ and which might be a limitation of the tracking performance, as the actual velocity measurement may be incorrect. As the radar utilizes this FMCW waveform, ambiguity issues in the range domain will not occur within the operational ranges in this research.

**Table 1.** Main characteristics of the PARSAX radar.

| Category | Parameter | Value |
|---|---|---|
| System characteristics | Center frequency ($f_c$) | 3.315 GHz |
| | Modulation bandwidth ($B$) | up to 50 MHz |
| | Sweep time ($T_s$) | 1 ms |
| | Effective bandwidth ($B_{eff}$) | up to 45 MHz |
| | Range resolution ($\Delta R$) | up to 3.3 m |
| Power characteristics | Maximum power per channel | 100 W |
| | Transmitter attenuation | up to 80 dB |
| Transmitter parabolic antenna | Antenna diameter | 4.28 m |
| | Antenna beamwidth | 1.8° |
| | Antenna gain | 40.0 dB |
| Receiver parabolic antenna | Antenna diameter | 2.12 m |
| | Antenna beamwidth | 4.6° |
| | Antenna gain | 32.8 dB |
| TX-RX isolation | HH-polarized | −100 dB |
| | VV-polarized | −85 dB |
| ADC characteristics | Maximum sampling frequency | 400 MHz |
| | ADC resolution | 14-bit |
| | Spur-free dynamic range | ≥70 dB |

The main challenge in designing a polarimetric radar is the orthogonality of the sounding signals, which strongly affects the isolation between the polarimetric radar channels. Bad polarimetric isolation limits the system's ability to observe weak targets in an environment with other strong targets and clutter. Therefore, a novel method for quasi-simultaneous measurement of the PSM was developed for the PARSAX radar [37]. Based on the work of Babur et al., a pair of *linear frequency modulated* (LFM) signals with a positive slope are time-shifted from each other to allow the occupation of different frequency bands continuously, as illustrated in Figure 1. For this research, a time delay that equaled 0.5 of the sweep repetition time was used. Due to this relative time delay corresponding to a frequency shift, the signals' orthogonality could be considered, and a high polarimetric isolation level could be achieved. Here, the first signal was transmitted with horizontal polarization, whereas the shifted LFM signal was vertically polarized. The FMCW deramping technique provided the possibility to mix each received signal with both transmitted signals, resulting in beat frequency spectra corresponding to all range bins and all scattering matrix elements. This provided the radar with the ability to transmit and receive horizontally and vertically polarized electromagnetic waves in parallel. Utilizing sounding signals with orthogonally polarized components is a strong advantage compared with many other existing polarimetric radar systems, which measure the elements of the polarization scattering matrix by switching the polarization mode of the transmitter or receiver from pulse to pulse. This causes a temporal measurement mismatch, which is not desirable when using polarization information for target detection and classification algorithms [34,36]. A detailed block diagram of the system of the PARSAX radar can be found in [36].

In order to measure the PSM of a target accurately, the measurements of the relative amplitudes and phases of the backscattered signals need to be accurate. This accuracy can be validated and improved with the radar's polarimetric calibration. In the case of the PARSAX radar, the polarimetric channels can be calibrated using two approaches. Internal and semi-external calibrations have been performed using the internal calibration circuits and cross-validation between simulations and measurements [38]. The external calibration of the radar system using a rotatable dihedral corner reflector has been conducted as well [39].

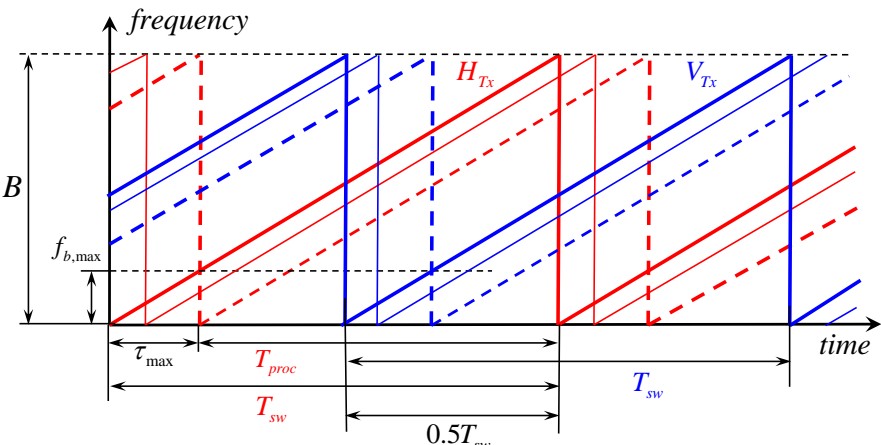

**Figure 1.** A pair of time-shifted LFM signals transmitted with horizontal and vertical polarization, illustrated in the chirp frequency domain (with the sounding signals indicated by solid thick red and blue lines, the received signals by thin solid lines, and the received signals with the maximum delay/distance from the radar by dashed lines). More details can be found in [37].

### 2.2. Data Acquisition

In this work, real-world polarimetric radar data were captured by PARSAX while observing a dense highway (the A13 between Delft and Rotterdam in The Netherlands). The targets were present at a slant range from approximately 3 km to 4 km. The radar was positioned on top of a building with a height of approximately 90 m and was pointed toward the highway. A map is visualized in Figure 2. As can be seen, the direction of the vehicles on the highway was approximately in the same direction as the line of sight of the radar. Hence, the radial velocity would almost be equivalent to the absolute velocity. Note that the maximum speed on this highway was 100 km h$^{-1}$.

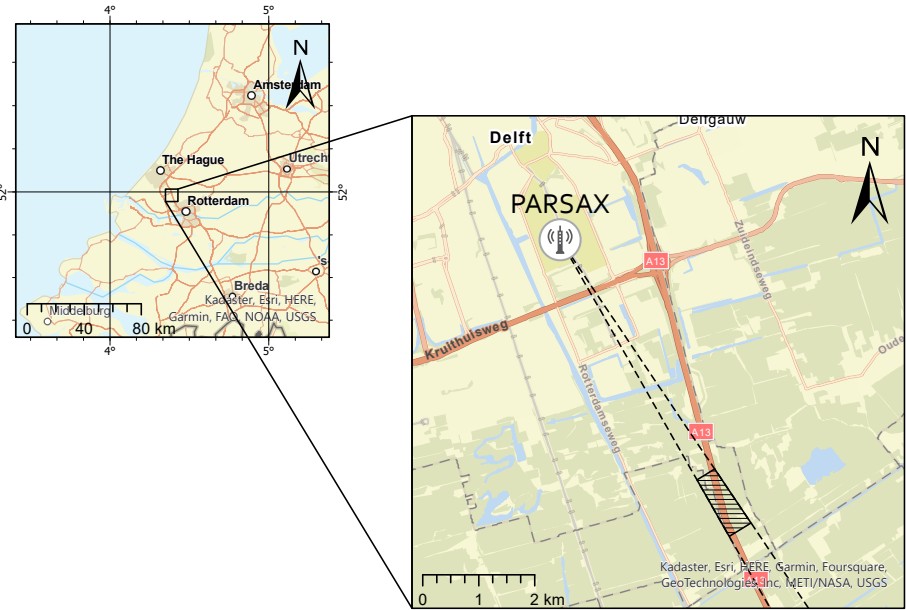

**Figure 2.** A map illustrating the location of the PARSAX radar and the illuminated area. The area of interest used for this analysis is indicated by black lines.

### 2.3. Signal and Data Processing Chain

The main concept of the $H/A/\alpha$ decomposition technique is that the $H$, $A$, and $\alpha$ features of the targets can be used to identify the underlying average scattering mechanism.

These parameters are extracted from eigenvalue analysis of the average coherency matrix **T** [19], which can be derived by either time averaging or spatial averaging. Hence, it is required to find clusters of detected cells that represent the targets and to track each target over time. Therefore, all moving vehicles that can be observed by the radar need to be detected and tracked, which is performed in the most straightforward manner in the range-Doppler domain. As a result, a dedicated signal and data processing chain to process the real-world data were developed, of which an overview is illustrated in Figure 3.

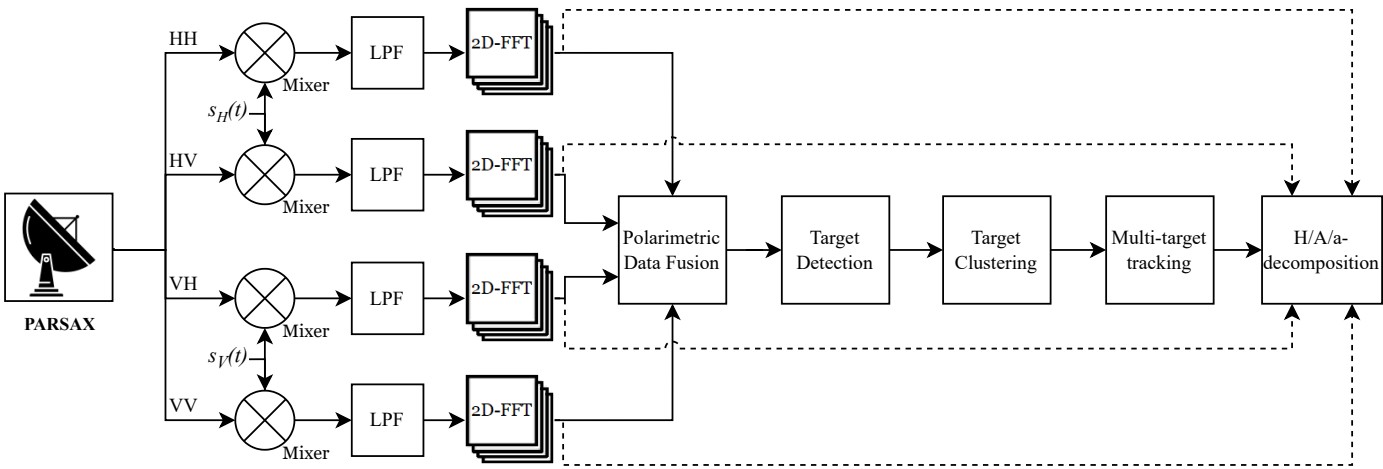

**Figure 3.** Signal and data processing chain for the four polarization receiver channels, where $s_H(t)$ and $s_V(t)$ represent the transmitted waveforms used for deramping the received signals for further processing.

### 2.3.1. Range-Doppler Processing

As target detection would be applied in the range-Doppler domain, a two-dimensional *fast Fourier transform* (FFT) was performed over fast and slow times to provide the range-Doppler map for each of the four polarimetric channels. Whereas the first FFT was applied for range compression with a Hamming window, before the second FFT, an 80 dB Chebyshev window function was applied to the data in order to reduce the sidelobe levels and spectral leakage such that the sidelobe level was constant [1]. The normalized range-Doppler spectrum of the first frame of a single polarization channel is shown in Figure 4, where the Doppler frequency was converted to the radial velocity.

As can be seen, the moving vehicles were indicated by peaks at a certain range and a certain velocity, which could either be negative (approaching the radar) or positive (receding from the radar). Beforehand, the sampled IF radar signals were preprocessed, and in order to get rid of reflections of the static clutter, which can clearly be seen around $0\,\mathrm{km\,h^{-1}}$, a sixth-order Butterworth *high-pass filter* (HPF) with its cut-off frequency at 122 Hz, equivalent to $20\,\mathrm{km\,h^{-1}}$, was applied [1]. From prior knowledge of the area of interest, it was assumed that the targets were not moving orthogonal to the radar view angle such that the radial velocity of the moving vehicles was near $0\,\mathrm{km\,h^{-1}}$ and filtered out as well. After the Doppler processing steps of 512 pulses per time frame, the strong reflections of the moving vehicles can clearly be seen in the range-Doppler map, as shown in Figure 5.

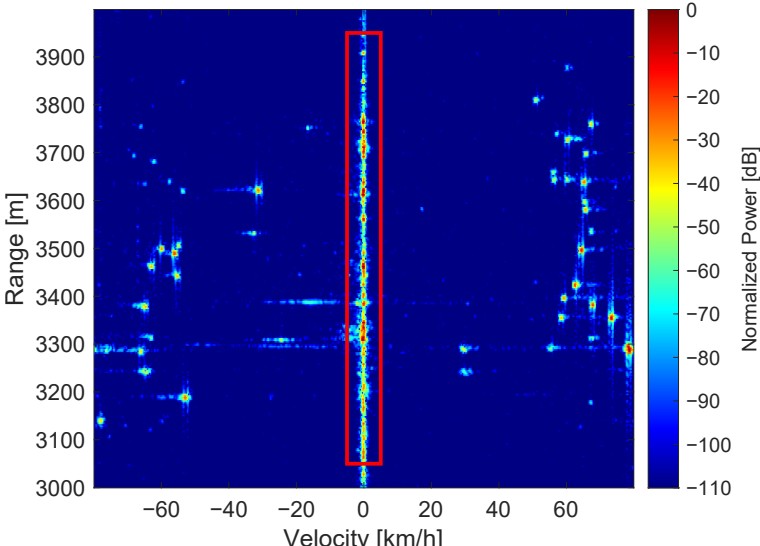

**Figure 4.** A range-Doppler map of real-world data representing a dense highway, visualizing moving vehicles and static clutter. This map originated from the HH channel, the other polarization channels (HV, VH, and VV) showed similar results. The red box around zero Doppler velocity shows the area of analyzed clutter signals.

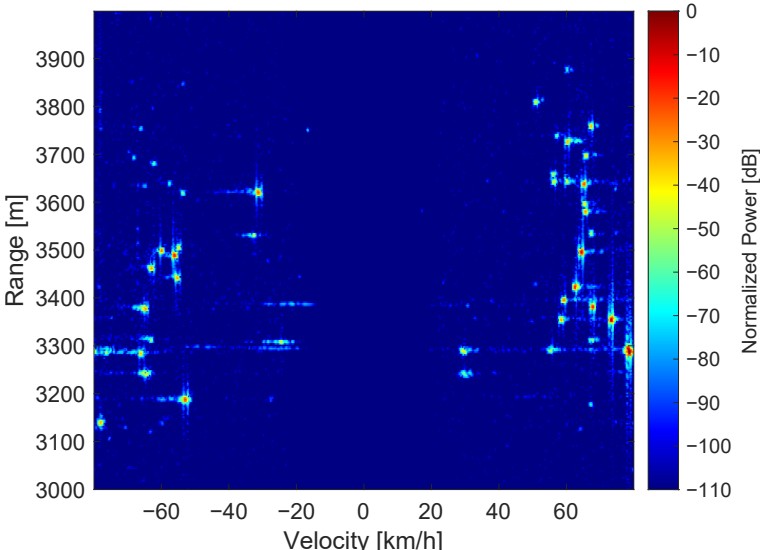

**Figure 5.** A range-Doppler map of real-world data representing a dense highway, visualizing only moving vehicles. This map originated from the HH channel, and the other polarization channels (HV, VH, and VV) showed similar results.

### 2.3.2. Polarimetric Data Fusion

Before target detection took place, to take full advantage of the additional polarimetric information, all four data sets were exploited by a data fusion method to improve the detection performance. All four range-Doppler spectra, corresponding to the four elements of the PSM, were integrated into a single range-Doppler map per time frame.

One approach of data fusion is the so-called *polarimetric matched filter* (PMF), which maximizes the target-to-clutter ratio. This method creates a detection map by fusing the data and applying a detection algorithm sequentially [40]. Other techniques exist that combine the data fusion and detection algorithm processing steps. For instance, the *optimal polarimetric detector* (OPD) incorporates the polarimetric information in the *likelihood ratio test* (LRT) [41]. Unfortunately, since both the PMF and the OPD rely on the assumption that the polarimetric characteristics of both the target and the clutter are known, they are often

not applicable to real-life scenarios. However, other suboptimal detectors that only require the characteristics of the clutter exist as well. One of these detectors is the *polarization whitening filter* (PWF), which minimizes the ratio of the standard deviation and the mean of the signal amplitude. This is achieved by using the (estimated) covariance matrix of the clutter in the test statistic for the LRT [42]. Other examples of a full polarimetric detector are the *span detector* (SD) [13] and the *polarimetric maximization synthesis detector* (PMSD) [14], which both do not require any a priori information. These approaches are based on a non-coherent summation of all polarization elements as a test statistic for the LRT, denoted as $\Lambda$, which are defined by

$$\Lambda(\mathbf{S}) = |S_{HH}|^2 + |S_{HV}|^2 + |S_{VH}|^2 + |S_{VV}|^2, \tag{1}$$

and

$$\begin{aligned} \Lambda(\mathbf{S}) = &\frac{1}{2}\left(|S_{HH}|^2 + |S_{HV}|^2 + |S_{VH}|^2 + |S_{VV}|^2\right) + \\ &\frac{1}{2}\sqrt{\left(|S_{HH}|^2 - |S_{VV}|^2\right)^2 + 4\left|S_{HH}^* S_{HV} + S_{VV} S_{VH}^*\right|^2}, \end{aligned} \tag{2}$$

respectively, where $\mathbf{S}$ represents the PSM of a single range-Doppler bin. The PMSD has been proposed as an improvement with respect to the SD.

By comparing several polarimetric fusion methods, Novak et al. [43] showed that the PMSD exhibits the highest detection performance without any a priori information about the polarization characteristics of the targets or clutter. Therefore, a fusion method based on the PMSD was implemented.

### 2.3.3. Target Detection and Clustering

Target detection means deciding for each range-Doppler bin whether a target is present or not. Often, this decision is based on a threshold with respect to the amplitude or power measurement of the range-Doppler bin. This threshold can either be a fixed, non-adaptive amplitude value or an adaptive threshold based on the estimation of the local noise level, which is incorporated in the commonly-used *constant false alarm rate* (CFAR) detector. To detect the moving vehicles in the range-Doppler domain, a non-adaptive detector, based on a fixed threshold and a fixed probability of false alarms $P_{FA}$, was compared with the *cell-averaging* (CA) CFAR detector, an adaptive detection method that is commonly used in radar applications which locally estimates the average noise level of each range-Doppler bin and compares it with the received power [44]. Extensions of this detector, such as the *ordered statistics* (OS) CFAR [45], the *smallest of cell-averaging* (SOCA) CFAR, and the *greatest of cell-averaging* (GOCA) CFAR [46], were compared among each other as well. Since the OS-CFAR is more robust to noise fluctuations and has shown a higher detection performance in a dense target environment [1], such as a highway, this detector was selected for further processing. Input parameters such as the $P_{FA}$, the number of training cells $N_t$, the number of guard cells $N_g$, and the rank were determined by a tuning process.

One disadvantage of this detection algorithm is its limiting the performance of fast-moving targets. The moving vehicles of interest have a few range cells of migration within the coherent processing interval. Although compensation of the range migration might increase the *signal-to-noise ratio* (SNR) of the target, this was not included in the processing chain for simplicity reasons. In future work, this CFAR detector could be replaced by a coherent CFAR detector dedicated to the detection of fast-moving targets in a high-range resolution mode, as proposed in [47]. Note that this shift (which is related to its phase shift) would be the same for both polarizations. Thus, the phase difference between the polarization channels would not change.

Moreover, applying this detector to a range-Doppler map results in a binary detection map with clusters of hits that represent the vehicles moving with a certain velocity at a certain range. These binary detection maps still contain some imperfections and distortions due to noise, clutter, and the limitations of the target detection algorithm. For example, clusters of detection cells often do not represent the target correctly due to holes and protruded

cells. Aside from that, clusters with relatively small extents are probably false alarms and do not represent targets. These imperfections are resolved by applying a morphological filter [48], exploiting the aliasing property of Doppler processing and discarding detections with low velocity with an additional HPF. Subsequently, the *density-based spatial clustering of applications with noise* (DBSCAN) algorithm is applied for clustering such that a group of detected cells represents a single target [49], which will simplify the tracking algorithm significantly.

As input for the OS-CFAR detector, the probability of a false alarm $P_{FA}$ was set to $10^{-8}$, the number of guard cells $N_g$ was set to $[6, 4]$, and the number of training cells $N_t$ was set to $[6, 6]$. An example of the resulting target detection and clustering performance can be seen in Figure 6.

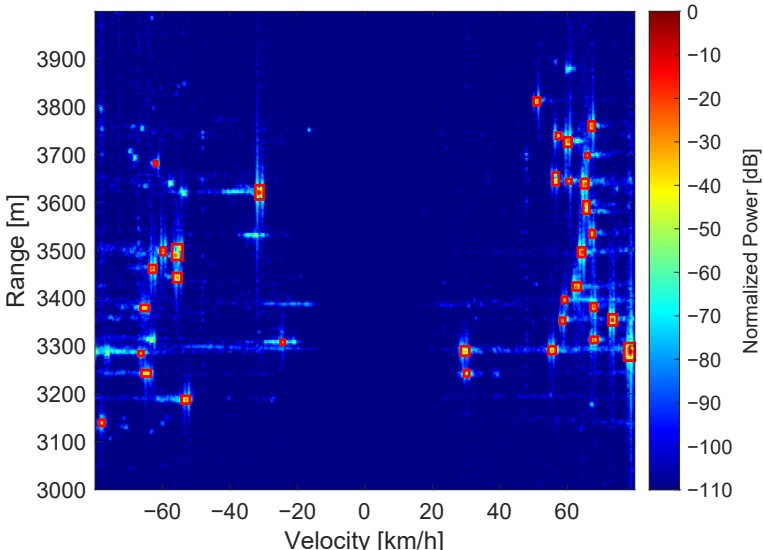

**Figure 6.** Resulting detection map and clusters (indicated by red boxes) after applying polarimetric data fusion and the OS-CFAR detector.

### 2.3.4. Multi-Target Tracking

To describe the polarimetric signatures of multiple different targets, each target needed to be tracked over time. In order to track multiple targets from frame to frame, a *multi-target tracking* (MTT) algorithm was developed. Each MTT problem is different, depending on the number of objects that are present, the sparsity of the objects, and the required performance of the tracking algorithm. Uncertainty concerning the observations or detections of the objects and the prediction of the objects' paths make a multi-target tracking problem challenging. A block diagram of the basic elements of an MTT algorithm [50] is shown in Figure 7.

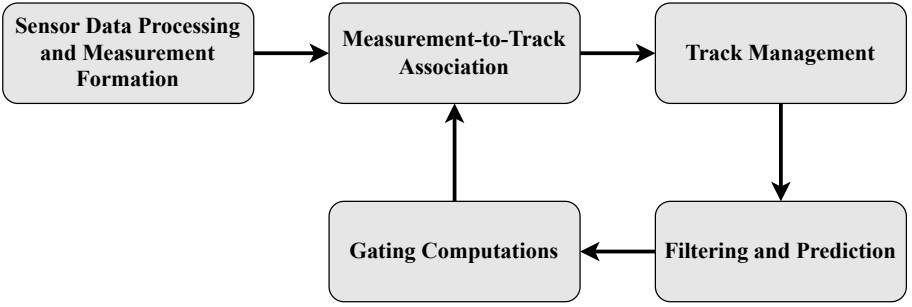

**Figure 7.** A block diagram of the basic elements of an MTT algorithm.

For tracking purposes, it was assumed that each target was represented by a single state per cluster found in the detection map, although it consisted of multiple detected cells.

The state of each *i*th target at the *k*th time frame $\mathbf{x}_k^i$ was described by its true range and true velocity. Based on the clusters found in the detection map of the fused range-Doppler frame, each target *i* was represented by a single noisy measurement $\mathbf{z}_k^i$ in a time frame *k*. Each measurement could be described by the observed states, which included a range value *R* and a velocity value *v* and were selected by the state observation matrix $\mathbf{H}$, following

$$\mathbf{z}_k^i = \mathbf{H}\mathbf{x}_k^i + \mathbf{v}_n, \tag{3}$$

where $\mathbf{v}_n \sim \mathcal{N}(0, \mathbf{\Sigma}_n)$ represents the zero-mean Gaussian measurement noise with noise variances $\sigma_{n,R}^2$ and $\sigma_{n,v}^2$ on the diagonal of $\mathbf{\Sigma}_n$.

A classic Kalman filter was implemented to update and predict the state of each target, following the constant acceleration state dynamics model. This model was selected based on its accurate approximation of the behavior of moving vehicles on a highway, and it is well-suited for tracking purposes in the range-Doppler domain. With this model, each state could be described by the target's range, velocity, and acceleration. This provided the following state transition model matrix $\mathbf{F}$, state observation matrix $\mathbf{H}$, and process noise covariance matrix $\mathbf{W}$:

$$\mathbf{F} = \begin{bmatrix} 1 & T & T^2/2 \\ 0 & 1 & T \\ 0 & 0 & 1 \end{bmatrix},$$

$$\mathbf{H} = \begin{bmatrix} 1 & 0 & 0 \\ 0 & 1 & 0 \end{bmatrix}, \tag{4}$$

$$\mathbf{W} = \begin{bmatrix} T^4/4 & T^3/2 & T^2/2 \\ T^3/2 & T^2 & T \\ T^2 & T & 1 \end{bmatrix},$$

where *T* is the time step between successive range-Doppler frames.

First, the target's predicted state $\hat{\mathbf{x}}_{k|k-1}$ would be estimated based on its previous state $\hat{\mathbf{x}}_{k-1|k-1}$. That aside, the predicted state covariance matrix $\hat{\mathbf{Q}}_{k|k-1}$ was computed based on its previous state covariance matrix $\hat{\mathbf{Q}}_{k-1|k-1}$ and the process noise covariance matrix $\mathbf{W}$. This is described by

$$\hat{\mathbf{x}}_{k|k-1} = \mathbf{F}\hat{\mathbf{x}}_{k-1|k-1},$$
$$\hat{\mathbf{Q}}_{k|k-1} = \mathbf{F}\hat{\mathbf{Q}}_{k-1|k-1}\mathbf{F}^T + \sigma_w^2\mathbf{W}, \tag{5}$$

where $\sigma_w^2$ is the process noise variance. Subsequently, based on the gain $\mathbf{K}_k$ computed by the predicted covariance matrix $\hat{\mathbf{Q}}_{k|k-1}$, the target's current state $\hat{\mathbf{x}}_{k|k}$ and the target's current state covariance matrix $\hat{\mathbf{Q}}_{k|k}$ would be estimated as follows:

$$\mathbf{K}_k = \hat{\mathbf{Q}}_{k|k-1}\mathbf{H}^T\left(\mathbf{H}\hat{\mathbf{Q}}_{k|k-1}\mathbf{H}^T + \mathbf{\Sigma}_n\right)^{-1},$$
$$\hat{\mathbf{x}}_{k|k} = \hat{\mathbf{x}}_{k|k-1} + \mathbf{K}_k\left(\mathbf{z}_k - \mathbf{H}\hat{\mathbf{x}}_{k|k-1}\right), \tag{6}$$
$$\hat{\mathbf{Q}}_{k|k} = (\mathbf{I} - \mathbf{K}_k\mathbf{H})\hat{\mathbf{Q}}_{k|k-1},$$

where $\mathbf{I}$ is a $3 \times 3$ identity matrix [50].

To simplify the data association problem, a hyperellipsoid gate which incorporated the statistical Mahalanobis distance $d_M$ was applied. Only measurements with a minimum likelihood that they could originate from the object corresponding to the track would be considered to be within the gate. For measurement *i* and track *j*, this gate condition was then described by

$$\left[\mathbf{z}_k^i - \mathbf{H}\hat{\mathbf{x}}_{k|k-1}^j\right]^T \mathbf{S}_{j,k}^{-1}\left[\mathbf{z}_k^i - \mathbf{H}\hat{\mathbf{x}}_{k|k-1}^j\right] \leq d_M, \tag{7}$$

where $\mathbf{S}_{j,k}$ is the covariance matrix of track $j$ at time frame $k$. This covariance matrix combines the predicted covariance of the target state $\hat{\mathbf{Q}}_{k|k-1}$ and the covariance of the measurement uncertainty $\mathbf{\Sigma}_n$, which is then given by

$$\mathbf{S}_{j,k} = \mathbf{H}\hat{\mathbf{Q}}^{j}_{k|k-1}\mathbf{H}^{T} + \mathbf{\Sigma}_n, \tag{8}$$

which is exactly the denominator of the Kalman gain calculation in Equation (6) [50].

To associate each measurement to a certain target, the *global nearest neighbor* (GNN) method, based on minimizing the total distance, was implemented. A cost matrix $\mathbf{C}$ containing this Mahalanobis distance between all measurements within the gate and each track, denoted by $C_{i,j}$, could be used to find the optimal combination of measurement-to-track pairs that minimized the total cost [1]. Given that each measurement could only be assigned once, and each track could only be associated with at most one measurement, this method could mathematically be described as follows:

$$
\begin{aligned}
\min \quad & \sum_{i=1}^{N_{meas}} \sum_{j=1}^{N_{tracks}} C_{i,j} \cdot z_{i,j} \\
\text{s.t.} \quad & C_{i,j} \leq d_M \\
& \sum_{i=1}^{N_{tracks}} z_{i,j} = 1, \quad \forall j \\
& \sum_{j=1}^{N_{meas}} z_{i,j} = 1, \quad \forall i
\end{aligned}
\tag{9}
$$

where $d_M$ is the Mahalanobis distance and where $z_{i,j}$ is equal to one when the measurement $i$ is assigned to track $j$; otherwise, it is equal to zero.

Due to the limitations of Doppler processing, ambiguity in the Doppler domain can occur. In some cases, the target's velocity measurement might not be correct, resulting in an incorrect state prediction and thus decreased tracking performance. In order to mitigate this problem, a *multiple hypothesis tracking* (MHT)-based approach was introduced, based on the work of Li et al. [51]. For each track (i.e., each vehicle), the likelihood ratio of two hypotheses for whether the Doppler frequency (proportionally related to the vehicle's radial velocity) of a new track was folded or not was computed to determine the next predicted state in the consecutive range-Doppler frames. To improve the computational complexity of the MHT algorithm, it was assumed that Doppler folding could only occur once.

Lastly, an M/N logic test was selected to solve the track management problem such that all unassigned measurements would initialize a new track, and all tracks that were not associated with a measurement would be canceled [52].

### 2.4. H/A/α Decomposition

While tracking all moving targets, the amplitude and phase information of all four polarization channels would be collected. This polarimetric information could be applied to polarimetric decomposition techniques, which could provide more information and a better insight into the physical meaning of the scattering of the moving vehicles [30]. Under the assumption of the monostatic backscattering case, the reciprocity rule applied (i.e., $S_{HV} = S_{VH}$). Therefore, to characterize the target scattering, the PSM of each range-Doppler bin could be represented by the Pauli scattering vector $\tilde{\mathbf{k}}_{\mathbf{P}}$, described by

$$\tilde{\mathbf{k}}_{\mathbf{P}} = \frac{1}{\sqrt{2}} \begin{bmatrix} S_{HH} + S_{VV} \\ S_{HH} - S_{VV} \\ S_{HV} + S_{VH} \end{bmatrix}. \tag{10}$$

The average coherency matrix **T**, based on the statistical average of all the scattering information, is then defined by

$$\mathbf{T} = \left\langle \tilde{\mathbf{k}}_{\mathbf{P}} \cdot \tilde{\mathbf{k}}_{\mathbf{P}}^{H} \right\rangle, \tag{11}$$

where $[\cdot]^{H}$ denotes the conjugate transpose operation and $\langle \cdot \rangle$ is used to describe the averaging process. As these decomposition techniques aim to provide an interpretation of the scattering mechanism, it was assumed that the average target scattering was invariant to polarization changes. Therefore, averaging could be performed in any domain (e.g., time domain, spatial domain, or angular domain).

Within the $H/A/\alpha$ decomposition method, the polarimetric entropy $H$ is a quantitative measure of randomness in the scattering mechanisms, where the anisotropy $A$ is used to characterize the scattering phenomenon and the alpha angle ($0° \leq \alpha \leq 90°$) represents the surface scattering characteristics, from isotropic surface scattering to dipole scattering and dielectric dihedral scattering [19,30]. These parameters can be very useful for characterizing the scattering properties [6]. To compute these parameters, the average coherency matrix **T** is decomposed as follows:

$$\mathbf{T} = \mathbf{U}\mathbf{\Lambda}\mathbf{U}^{H} = \mathbf{U} \begin{bmatrix} \lambda_1 & 0 & 0 \\ 0 & \lambda_2 & 0 \\ 0 & 0 & \lambda_3 \end{bmatrix} \mathbf{U}^{H}, \tag{12}$$

where **U** is a unitary matrix containing three orthogonal eigenvectors $\mathbf{u}_i$ and $\mathbf{\Lambda}$ is a $3 \times 3$ diagonal matrix with nonnegative real eigenvalues ($\lambda_1 \geq \lambda_2 \geq \lambda_3 \geq 0$). If all eigenvalues $\lambda_i$ are zero, except for $\lambda_1$, then the coherency matrix **T** represents a single scattering matrix, corresponding to a pure correlated and completely polarized scattering mechanism. On the other hand, if all eigenvalues are identical, then **T** corresponds to a completely unpolarized and random scattering process [6].

To define the polarimetric entropy $H$ and alpha angle $\alpha$, the pseudo-probabilities $P_i$ need to be obtained from the eigenvalues $\lambda_i$ ($i = 1, 2, 3$), which can be described as follows:

$$P_i = \frac{\lambda_i}{\sum_{j=1}^{3} \lambda_j}. \tag{13}$$

The $H/A/\alpha$-decomposition is then defined by

$$H = -\sum_{i=1}^{3} P_i \log_3(P_i), \tag{14}$$

$$A = \frac{\lambda_2 - \lambda_3}{\lambda_2 + \lambda_3}, \tag{15}$$

$$\alpha = \sum_{i=1}^{3} P_i \cos^{-1}(|u_{i,1}|), \tag{16}$$

where $u_{i,1}$ is first element on the eigenvector $\mathbf{u}_i$ [19]. These equations show that the polarimetric entropy is $H = 0$ for a completely deterministic scattering mechanism and $H = 1$ for a completely random scattering mechanism. A completely deterministic scattering mechanism, meaning that the scattering wave is completely polarized, results in a degree of polarization of one. Correspondingly, completely random scattering (i.e., $H = 1$) leads to a degree of polarization of zero. In the latter case, the process is completely depolarizing, and polarimetric information is useless for classification [6].

## 3. Results

The proposed signal and data processing chain (see Figure 3) was applied to real-world polarimetric radar data. Beforehand, the performance of the MTT algorithm was analyzed. This was investigated by applying the proposed signal and data processing chain

to simulated range-Doppler data. Subsequently, the processing chain was applied to the real-world data, and the $H/A/\alpha$ decomposition technique was utilized to describe the polarimetric signatures of the moving automotive vehicles.

### 3.1. Tracking Performance

To test our approaches, verify our assumptions, and validate the implementation of the proposed MTT algorithm, a simulation of the range-Doppler data was implemented. A set of binary detection maps for 30 time frames was synthesized as the input for the tracking algorithm. In this simulation, 20 targets were created, of which the initial range $R_0$ and initial velocity $v_0$ were uniformly distributed over the range from 3300 m to 3400 m and over velocities from 60 km h$^{-1}$ to 100 km h$^{-1}$ and from $-60$ km h$^{-1}$ to $-100$ km h$^{-1}$. Its acceleration was initialized by following a normal distribution with a mean of 0 and a standard deviation of 0.1. Each target's true state was updated according to a dynamic model while assuming constant acceleration. Additive white Gaussian noise was added to the target's centroid at each time frame, resulting in a noisy track of a point target in the range-velocity domain. In order to mimic real-world targets, each point target was dilated with a disk-shaped element. The simulation results are presented in Figure 8, showing the true trajectory and output of the tracking algorithm.

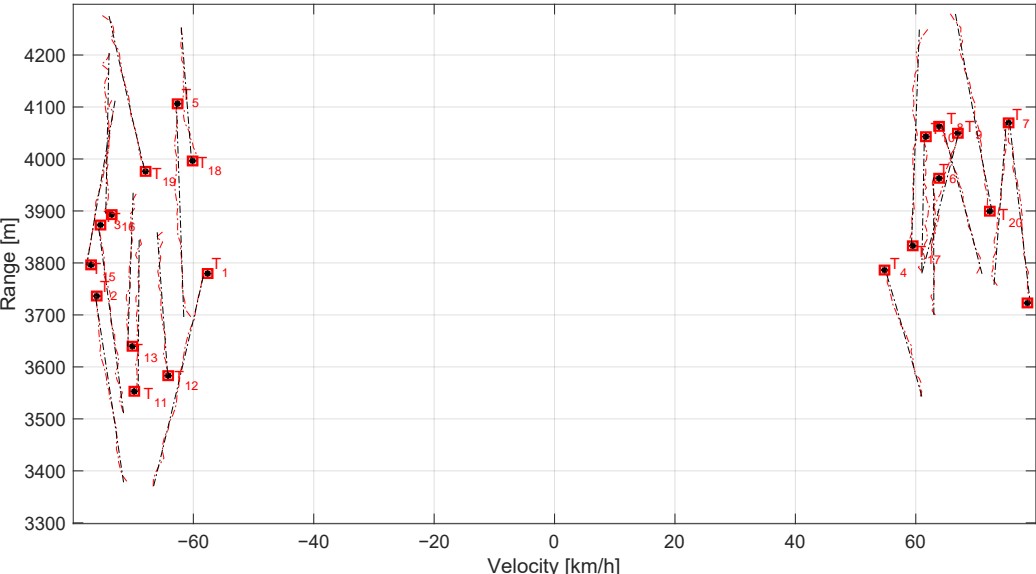

**Figure 8.** Synthesized trajectories (indicated in black) and their corresponding estimations (indicated in red) as a result of the proposed multi-target tracking algorithm in the range-velocity domain of $N_T = 20$ targets for $T_{sim} \approx 15$ s.

As can be seen, even in a dense environment, the tracking algorithm was able to cope with closely spaced measurements and tracks crossing each other. As an MHT-based approach was introduced to handle the Doppler ambiguity, approximately 50% of the simulated targets followed a track with measurements corresponding to a folded velocity. This simulation was used to verify the functionality of this approach and to analyze the improvements of the algorithm when dealing with Doppler ambiguity. To examine the tracking performance, the state dynamics over time of the estimated track of the target $T_5$ would be analyzed. Note that this target was initialized with an absolute velocity greater than the maximum unambiguous velocity $v_r^{max}$, and thus its velocity measurements were folded once. In Figure 9a,b, the true and estimated range and the *root mean square error* (RMSE) of the estimation (in terms of the range resolution $\Delta R$) are shown, respectively. Moreover, Figure 9c shows the velocity estimation in comparison with the target's true velocity along with its corresponding RMSE (in terms of the velocity resolution $\Delta v_r$) in Figure 9d.

It can be seen that the estimated range almost perfectly followed the true range. The RMSE of the estimation of the range and the velocity of the target with respect to its true range and velocity were found to be smaller than $0.5\,\mathrm{m}$ and $0.2\,\mathrm{m\,s^{-1}}$, respectively.

Subsequently, this MTT algorithm was applied to the real-world polarimetric radar data, as introduced in Section 2.2. When visually inspecting Figure 10, illustrating the track results after 20 time frames, for each target, the estimated trajectory in the range-velocity domain is shown. The estimated current state $\hat{\mathbf{x}}_{k|k}$, predicted state $\hat{\mathbf{x}}_{k|k-1}$, and corresponding associated measurement $\mathbf{z}_k^i$ are plotted as well.

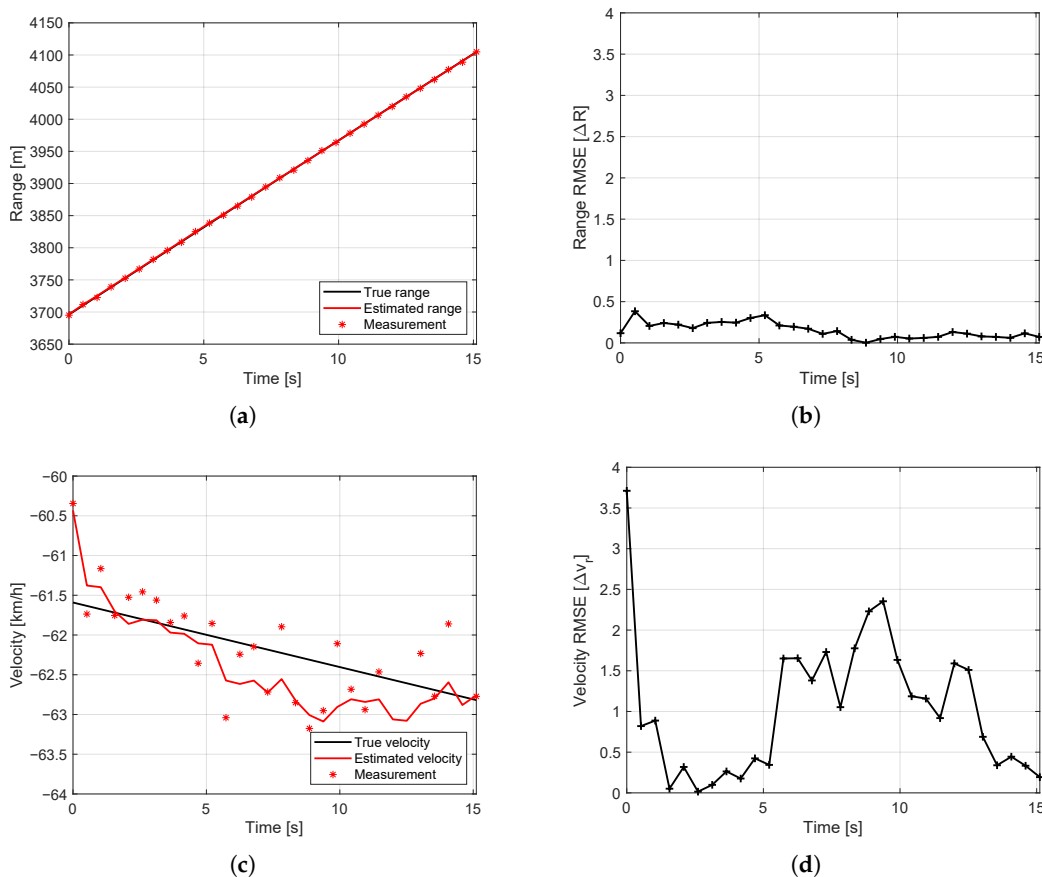

**Figure 9.** (**a**) The true and estimated range of target $T_5$ over time with (**b**) the corresponding RMSE, expressed in terms of the range resolution $\Delta R$, as well as (**c**) the true and estimated velocity of target $T_5$ over time with (**d**) the corresponding RMSE, expressed in terms of the velocity resolution $\Delta v_r$.

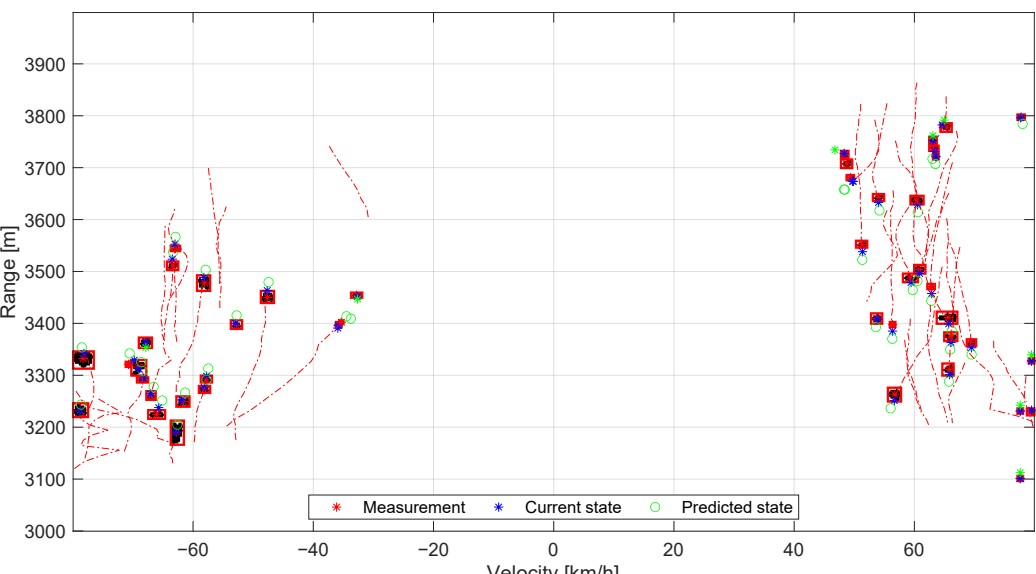

**Figure 10.** Estimated trajectories in the range-velocity domain of vehicles on a highway after 20 time frames. For each target, the estimated trajectory, estimated current state $\hat{\mathbf{x}}_{k|k}$, predicted state $\hat{\mathbf{x}}_{k|k-1}$, and corresponding associated measurement $\mathbf{z}_k^i$ are plotted.

### 3.2. Analysis of Polarimetric Signatures

While tracking each target, the amplitude and phase information of all four polarization range-Doppler maps, cluster information, target size, and target dynamics would be collected. As a result, the polarimetric information of 66 targets in a period of 46.5 s, representing 93 time frames, was acquired. As soon as the PARSAX radar had a range resolution of a few meters, the targets' polarimetric response would be affected by the multi-path effects within the range resolution volume, as the received signal equaled the sum of many signals (radar-car-radar, radar-road-car-radar, radar-road-car-road-radar, etc.). In this study, we used the measured polarimetric responses as they were measured, without any attempts to distinguish the components of these multi-path integral signals. The $H/A/\alpha$ decomposition technique was utilized to describe the polarimetric signature of the moving automotive vehicles. Describing the scattering mechanism based on the entropy $H$ and $\alpha$ angle is often performed in a two-dimensional $H/\alpha$ plane [19]. To characterize the scattering of the vehicles, the average coherency matrix **T** needed to be defined for each target. Therefore, averaging was required (see Equation (11)), which for this application could either be time averaging or spatial averaging.

Aside from that, to compare the characteristics of the targets, the polarimetric information of the static clutter was collected. This static clutter was the polarimetric data within the zero-Doppler region, which was filtered out during preprocessing and originated from the same area as the highway region with the targets, as indicated in Figure 4. This represents various sources of clutter, such as grass fields, trees, plants, and road infrastructure. As the cross-range of the area was approximately 100 m (for a given beamwidth, as pointed out in Table 1), grass fields and road infrastructure were the main contributors to the clutter.

In order to perform time averaging to find the average coherency matrix $\mathbf{T}_{time}$, the PSM of the target's centroid bin of each time frame was collected and averaged over time, resulting in a single value for $H$, $A$, and $\alpha$ for each target. A two-dimensional histogram of the entropy $H$ and the angle $\alpha$ of 66 targets was visualized in the $H/\alpha$ plane, as shown in Figure 11a. That aside, the time-averaged polarimetric information of the static clutter is depicted in Figure 11b as a two-dimensional histogram in the $H/\alpha$ plane as well. The overlapping lines on top of the data can be used for classification purposes, which will be discussed in Section 4.

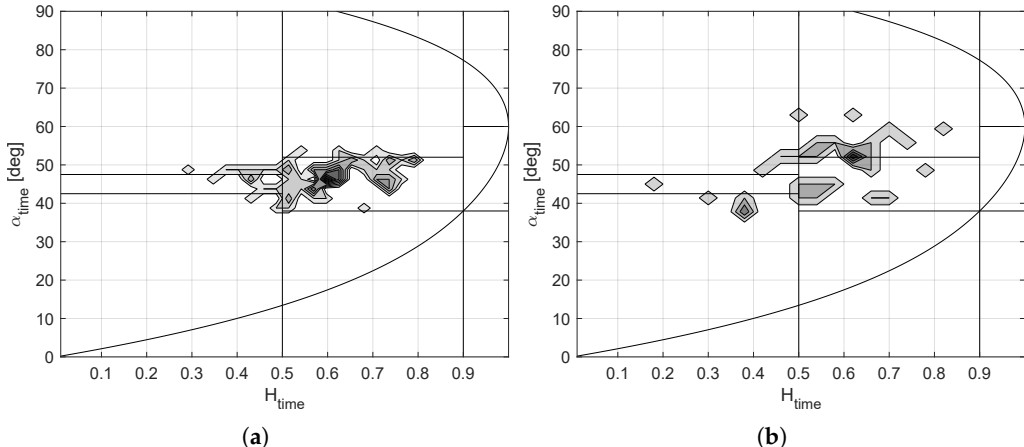

**Figure 11.** Two-dimensional histogram in the $H/\alpha$ plane, based on time averaging of the coherency matrix $\mathbf{T}_{time}$ of (**a**) moving automotive vehicles and (**b**) static clutter.

Spatial averaging is implemented to find the average coherency matrix $\mathbf{T}_{space}$. In this case, the PSM of all bins of each target's cluster was collected and averaged, resulting in a single value for $H$, $A$, and $\alpha$ for each target for each time frame. This resulted in 93 values for $H$, $A$ and $\alpha$ per target. Again, the two-dimensional histogram of all targets of all time frames is visualized in the $H/\alpha$ plane in Figure 12a. Similarly, spatial averaging was applied to the polarimetric information of the static clutter as well. The resulting distribution is displayed in Figure 12b.

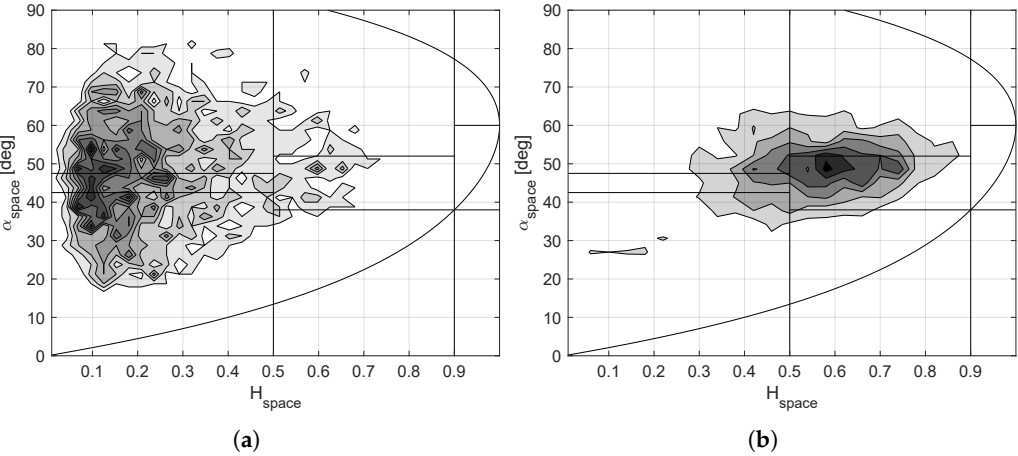

**Figure 12.** Two-dimensional histogram in the $H/\alpha$ plane, based on spatial averaging of the coherency matrix $\mathbf{T}_{space}$ of (**a**) moving automotive vehicles and (**b**) static clutter.

## 4. Discussion

Classification based on polarimetric signatures is often performed in the two-dimensional $H/\alpha$ plane. Since the anisotropy $A$ only becomes an important and useful parameter to describe the scattering process when the entropy $H$ is very high, this value was not incorporated in this analysis. This $H/\alpha$ plane could be separated into nine classes of basic scattering mechanisms, as presented in [19] by Cloude. This classification scheme illustrates a simple schematic overview to classify and describe the scattering mechanism based on typical values for the entropy $H$ and angle $\alpha$.

As can be seen in Figure 11, the entropy of the polarimetric signatures of the moving vehicles is distributed in the region $0.3 \leq H_{time} \leq 0.8$ and around $\alpha_{time} \approx 45°$. According to the classification schematic, this is correlated with the volume diffusion and quasi-deterministic or moderately random entropy, corresponding to dipole and anisotropic

particle scattering. Since only the centroid had been taken into account, this is probably related to the scattering of the main body of a vehicle. In the figures, it can be seen that the distribution of the data in the $H/\alpha$ plane corresponding to static clutter showed many similarities with the distribution of the data corresponding to the moving vehicles. Hence, this approach was not considered useful for discriminating moving automotive vehicles from a static environment.

On the other hand, the polarimetric signature based on spatial averaging of the $H/A/\alpha$ decomposition technique (see Figure 12) showed that a high concentration in the region at $H_{space} \leq 0.3$ and $30° \leq \alpha_{space} \leq 70°$ is present, corresponding to quasi-deterministic entropy with all kinds of scattering mechanisms. Since the target's cluster covered reflections from all parts of the vehicle (the main body, wheels, car mirrors, etc.), there was an obvious explanation for this scattering behavior in the $H/\alpha$ plane. Additionally, all scattering points of the vehicles provided linearly polarized reflected signals. Meanwhile, when inspecting the two-dimensional histogram describing the polarimetric signature of the static clutter, the resulting distribution showed that there is a big difference in the scattering properties with respect to the moving vehicles. The static clutter scattered in both orthogonal polarizations due to relatively strong depolarization properties. Therefore, this averaging approach is very suitable for describing the polarimetric signatures of moving vehicles. This approach gave the opportunity to directly, without consideration of the motion of the targets, compare the polarization features of the moving targets and static clutter. Note that this was valid for the type of clutter that was present in the used data (i.e., grass fields and road infrastructure) and that other clutter types (e.g., bare soil and water) do not particularly follow this behavior.

For classification purposes, the two-dimensional histograms in the $H/\alpha$ plane of moving vehicles and static clutter were visualized in a feature space (see Figure 13). Here, the mean values for $H_{space}$ and $\alpha_{space}$ over time were used such that each data point represented a single target or single clutter point. This feature space confirmed that the features $H_{space}$ and $\alpha_{space}$ showed compact and well-separated clusters corresponding to the target features and clutter features. These features even allowed instant classification, as time averaging was not required to derive these parameters. Therefore, it can be concluded that with these features, even without velocity information, a classifier should be able to accurately distinguish moving vehicles from static clutter.

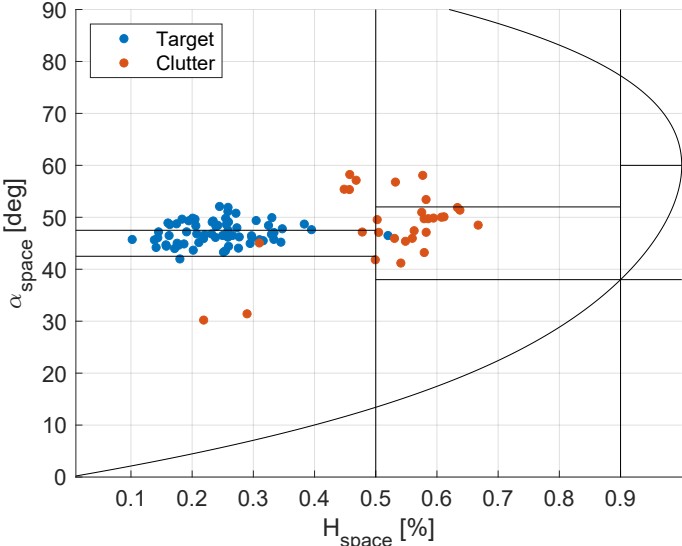

**Figure 13.** Feature space with the mean entropy $H$ against the mean angle $\alpha$, based on spatial averaging of the coherency matrix $\mathbf{T}_{space}$ of target scattering and static clutter scattering.

Although these results are very promising for detection and classification purposes, the following needs to be highlighted. As was explained in Section 2.1, the PARSAX system

measures all four elements of the PSM *quasi*-simultaneously. Due to the time shift between the pair of polarization-orthogonal LFM signals (see Section 2.1), the moving target would be displaced, resulting in a significant Doppler-proportional extra phase difference between the polarization channels, which could influence the results of the polarimetric feature analysis. However, it was not expected that this phase difference affected the polarimetric entropy $H$—the main polarimetric feature for this study—significantly, as it has a pure stochastic nature. Although the Doppler processing and the following averaging process partly solved this problem, in further studies on the statistics of specific polarimetric characteristics, the correction of this deterministic Doppler-dependent phase shift can be implemented quite easily and included in the feature extraction algorithm.

Furthermore, in this research, the polarimetric signatures were described in the two-dimensional $H/\alpha$ plane based on the $H/A/\alpha$ decomposition. Obviously, other polarimetric features of moving automotive vehicles can be extracted using alternative methods as well. For example, Ferrentino et al. studied the sensitivity of the polarimetric features extracted from dual-polarimetric SAR images. Classification of earthquake damage levels was carried out by exploiting a coherent dual-polarization feature based on the inter-channel coherence [53] and by exploiting the difference between two covariance matrices of two dual-polarized SAR images (before and after the earthquake) [54]. Another example of exploiting polarimetric features was given by Migliaccio et al. [55], where the experimental results showed that the entropy $H$ and the correlation between the co- and cross-polarized scattering amplitudes, based on the reflection symmetry theory [35], are relevant for detection and distinguishing oil slicks and man-made metallic targets from the sea's surface. These examples of polarimetric features are promising for the application of this analysis and can also be used for further research on the polarimetric signatures of moving automotive vehicles.

## 5. Conclusions

Motivated by the potential of the exploitation of polarimetric Doppler signatures for automotive target classification, a dedicated polarimetric radar signal and data processing chain for extracting the polarimetric features of moving targets was developed. The chain includes detection and tracking steps for moving vehicles in a multi-target environment in the range-Doppler domain. In this processing chain, the targets are detected by an OS-CFAR detector after application of polarimetric data fusion to take full advantage of the additional polarimetric information provided by the fully polarimetric radar. The developed processing chain was applied to real-world polarimetric radar data captured by the fully polarimetric S-band PARSAX radar while observing dense traffic on a highway. To cope with Doppler ambiguity of the available data, a dedicated multi-target tracking algorithm was implemented. The $H/A/\alpha$ decomposition technique was applied to the extracted target responses. The entropy $H$, anisotropy $A$, and angle $\alpha$ were used to describe the scattering characteristics and computed from the average coherency matrix **T**. By employing both time averaging and spatial averaging, the responses of both the targets and static clutter were visualized with two-dimensional histograms in the $H/\alpha$ plane. It was shown that the spatial averaging approach provides clear separation between moving automotive vehicles and static clutter. This approach gives the opportunity to directly, without consideration of the motion of the targets, compare the polarization features of moving targets and static clutter.

**Author Contributions:** Conceptualization, D.A.B. and O.A.K.; methodology, D.A.B.; software, D.A.B.; validation, D.A.B. and O.A.K.; formal analysis of results and discussion, D.A.B., O.A.K. and A.Y.; investigation, D.A.B. and O.A.K.; resources, D.A.B. and O.A.K.; data curation, O.A.K.; writing—original draft preparation, D.A.B.; writing—review and editing, D.A.B., O.A.K. and A.Y.; visualization, D.A.B.; supervision, O.A.K. and A.Y.; project administration, A.Y.; funding acquisition, A.Y. All authors have read and agreed to the published version of the manuscript.

**Funding:** This research received no external funding.

**Institutional Review Board Statement:** Not applicable.

**Informed Consent Statement:** Not applicable.

**Data Availability Statement:** The data presented in this study are available upon request from the corresponding author.

**Acknowledgments:** The authors would also like to thank Fred van der Zwan for his assistance with the measurements.

**Conflicts of Interest:** The authors declare no conflict of interest.

## Abbreviations

The following abbreviations are used in this manuscript:

| | |
|---|---|
| CA-CFAR | Cell averaging CFAR |
| CFAR | Constant false alarm rate |
| DBSCAN | Density-based spatial clustering of applications with noise |
| FFT | Fast Fourier transform |
| FMCW | Frequency-modulated continuous-wave |
| FPGA | Field-programmable gate array |
| GOCA-CFAR | Greatest of cell averaging CFAR |
| GNN | Global nearest neighbor |
| HPF | High-pass filter |
| LFM | Linear frequency modulated |
| LRT | Likelihood ratio test |
| MHT | Multiple hypothesis tracking |
| MTT | Multi-target tracking |
| OPD | Optimal polarimetric detector |
| OS-CFAR | Ordered statistics CFAR |
| PMF | Polarimetric matched filter |
| PMSD | Polarimetric maximization synthesis detector |
| PSM | Polarization scattering matrix |
| PWF | Polarization whitening filter |
| RCS | Radar cross-section |
| RMSE | Root mean square error |
| SAR | Synthetic aperture radar |
| SD | Span detector |
| SNR | Signal-to-noise ratio |
| SOCA-CFAR | Smallest of cell averaging CFAR |

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
