# Peer review of "An Advanced Data Processing Algorithm for Extraction of Polarimetric Radar Signatures of Moving Automotive Vehicles Using the H/A/α Decomposition Technique"

_remotesensing, doi:10.3390/rs15041060_

Round 1

Reviewer 1 Report

The ms in well-written and it contains interesting material for the radar-based automotive community.

I have some concerns that are listed as follows:

1. The detection of man-made targets can be easily performed using the inter-channel correlation. The underpinning hypothesis relies on the different symmetry properties resulting from natural and man-made targets. Please, refer to: 10.1109/JSTARS.2022.3217889 or 10.5589/m11-054 

those approaches should be lesse sensitive to the target's type

2. The use of the eigenvalue decomposition to extract polarimetric info from the measured covariance matrix is - in general - an interesting approach. However, I do not fully understand which is the physical ground that underpins the proposed methodology. Why should stationary target be distinguishable from moving one using eigenvalue parameters? I think that an improved theoretical analysis is needed to make the experimental discussion more robust. In fact, one can argue that the obtained results stem from a peculiar clutter environment that results in highly depolarizing backscatter. It is very simple to add a number of examples where the clutter does not follow this behavior (e.g.; bare soil, water, etc.). 

3. Following the previous comment, this Reviewer thinks that a detailed analysis of the clutter is needed. Which is the dominant clutter type? 

4. A straightforward approach to discriminate moving versus static targets is exploiting the Doppler shift. Why don't using it? Please, provide a discussion based on the intercomparison between Doppler-based analysis and eigenvalue-based analysis. This would double check your results and could be also useful to gain a better understanding of the physic that is at the basis of the approach you proposed.

Reviewer 2 Report

The paper is devoted to the study of polarimetric properties of moving targets using fully polarimetric Doppler surveillance S-band radar PARSAX. The instrument itself, the method for estimating full scattering matrix of moving target using orthogonal sounding signals, the results that could be obtained are of high interest. At the same time, there are some remarks that the authors should take into account.

1. I think the paper title reflects the content of the paper incompletely. Though the paper title refers to “Application of the H/A/ decomposition in an Advanced Data Processing Algorithm…”, only 5 pages of 20 are devoted to the topic declared in the title.

 2. I cannot agree with the statement "this radar has a maximum unambiguous range Rmax of approximately 153km" (see line 143), because in the scheme of polarimetric radar with two orthogonal LFM signals shifted in time at a half of sweep time the Rmax should be twice shorter.

3. In 2.3.2 authors mention various polarimetric filters and detectors considered in their processing. The paper would benefit if respective references were given there. The same about 2.3.3 (lines 217-224).

 4. Block-scheme of the signal and data processing chain in Fig.2 is incomplete. In current version, after 2D FFT we will get the “range frequency - Doppler” domain, not “range - Doppler”. I mean to say that the de-ramping or de-chirping stage (and LPF) must be shown somewhere before 2D FFT processing.

5. The length of coherent processing interval is equal to the burst length (512 sweeps). According to 0.5 s burst duration, the target moving with 20 m/s velocity will pass through ~3 range pixels in the case of 3 m range resolution. For a better signal to noise ratio a correction of the range migration for each moving target would be desirable.

6. T_shift value is not mentioned elsewhere in the paper under review, but if I am correct, it equals to the half of sweep time. At least, in ref 43 we can find an indication that tshift = T/2, where T is sweep time. Also, in ref 42 recommended t_shift = (0.4-0.6) T_sweep.

7. In scheme of polarimetric measurements accepted in the paper it is not correct yet to call the measurements of the scattering matrix elements as simultaneous ones because t_shift is non zero. In this scheme the measurements of the scattering matrix of moving targets will be distorted. The reason is that during time interval t_shift=0.5 ms the target moving with radial velocity 20 m/s will be displaced at 1 cm in range. As a result, 80 degs extra phase shift will be present in the phases of signals from the second column of the scattering matrix.

8. By the way, I could not find any information in journals and conference papers about polarimetric calibration of PARSAX system. So far, we have no idea about level of the instrument intrinsic polarimetric phase difference. What can be said about that?

9. Anyways, the phase discrepancy between first and second columns estimated for the targets with 20 m/s velocity corrupts identification of their real scattering mechanisms. The 80 degs polarimetric phase difference may be interpreted mistakenly as a result of combination of single scattering and double bounce scattering. I mean to say that H-alpha diagrams in the paper may be incorrect. In any case, the distribution of points on these diagrams is really different from what is typical in remote sensing of Earth covers by means of polarimetric SAR systems.

10. A problem of multipath propagation is not discussed (car-back, road-car-back, road-car-road-back…).

11. Where is the location of clutter area in the Range-Doppler domain used to estimate clutter properties? Would be nice to point it out.

12. Line 427: “The results related to the polarimetric signature of static clutter in the H/_-plane show many similarities with the polarimetric signature corresponding to moving vehicles” – paper reference is desirable.

 13. Finally, the statements in the Conclusions section regarding the interpretation of scattering mechanisms may be incorrect because of the phase errors mentioned in remarks 7 and 9.

Round 2

Reviewer 1 Report

I am happy with the ms as it stands now. No further action is needed

Reviewer 2 Report

I am really impressed with the job the authors did to work out our remarks and to improve the quality of their paper. I think there is no need to go through another cycle of comments and responses to comments.

I just want to make a couple of farewell remarks.

1. When authors tried to work out my remark about multipath propagation, they appealed to that fact all the signals distances are the same up to PARSAX resolution. But my concern was about “road-car-back” or double bounce propagation, when VV signal gets 180 deg extra phase shift compared with HH signal.

2. Authors still did not provide paper refs I was asking about in remark  N12.

Anyways, I do not insist on further corrections and propose to accept the current version of the paper as is.